# Spatial-Temporal Characteristic Analysis of Ethnic Toponyms Based on Spatial Information Entropy at the Rural Level in Northeast China

**DOI:** 10.3390/e22040393

**Published:** 2020-03-30

**Authors:** Fei Zhao, Yao Fu, Guize Luan, Sujin Zhang, Jingzhi Cai, Jieyu Ding, Jiangkang Qian, Zhiqiang Xie

**Affiliations:** School of Earth Science, Yunnan University, Kunming 650091, Yunnan, China; vwobai@163.com (F.Z.); fuyaoynu@163.com (Y.F.); luanguize@163.com (G.L.); zsj@mail.ynu.edu.cn (S.Z.); 18487211757@163.com (J.C.); uyeijgnid_7190@163.com (J.D.); qjk@mail.ynu.edu.cn (J.Q.)

**Keywords:** geographical distribution, Manchu culture, rural settlements migration, spatial information entropy

## Abstract

As a symbol language, toponyms have inherited the unique local historical culture in the long process of historical development. As the birthplace of Manchu, there are many toponyms originated from multi-ethnic groups (e.g., Manchu, Mongol, Korean, Hui, and Xibe) in Northeast China which possess unique cultural connotations. This study aimed to (1) establish a spatial-temporal database of toponyms in Northeast China using a multi-source data set, and identify their ethnic types and origin times; and (2) explore the geographical distribution characteristics of ethnic toponyms and the evolution of rural settlements by comparing the spatial analysis and spatial information entropy methods. The results found that toponyms reflect not only the spatial distribution characteristics of the density and direction of ethnic groups, but also the migration law of rural settlements. Results also confirm that toponyms contain unique cultural connotations and provide a theoretical basis for the protection and promotion of the cultural connotations of toponyms. This research provides an entropic perspective and method for exploring the spatial-temporal evolutionary characteristics of ethnic groups and toponym mapping.

## 1. Introduction

Toponyms not only represent geographical locations, but also represent spatial entities. Spatial scope is defined by geographical coordinates. In different time scales, in addition to their original meanings, there are also historical, cultural, social, and national meanings endowed by local residents. As carriers of geographic information, toponyms interpret the evolution of geographical environments at a small scale and play important roles as mediums in the research field [1,2]. This has such significant value for studying the history of a region that Fudan University and Harvard University collaborated together on the establishment of the China Historical Geographic Information System (CHGIS) project which aims to create a set of basic geographic information for the study of continuous changes in Chinese historical periods, as well as providing a GIS data platform, time statistics, searching tools, and models for related researchers [3,4,5,6]. As the birthplace of Manchu, Northeast China is home to Mongol, Han, Korean, Xibe, and other ethnic groups. As symbols of precious cultural heritage, the toponyms not only preserve the abundant language materials and accumulate a unique cultural connotation to a certain extent but can also reflect the local cultural characteristics, can serve as "living fossils" of the study of history and culture, and describe the history of all ethnic groups’ cultural blending and changes in rural settlements [7,8]. By identifying these unique toponyms remaining from the previous dynasties, analyzing the spatial distribution characteristics of toponyms and cultural landscape and analyzing the rural settlements built in different dynasties from the perspective of time, we can not only delineate the Manchu immigrant culture, but also reflect the geographical distribution over longer time scales.

The structure of spatial-temporal information involves many disciplines and application fields. The spatial-temporal distribution of meaningful variables can be discussed and analyzed from the perspective of decision making, and it can also be analyzed separately or jointly from the information viewpoint [9,10]. Some scholars, based on the concept of landscape ecology, have patched size distribution and permutation entropy into biomedical signal processing, from which the framework of spatial and temporal entropy analysis of a class of variables is derived. This framework, which can coordinate the classical methods related to entropy with the latest literature, and better consider the spatial-temporal embedding of information as well as the method of containing entropy, has been applied using land cover evolution data as an example [11]. At present, it is generally believed that map information needs to be measured. The application of information entropy in maps mainly focuses on the measurement of map spatial information such as the measurement of information of point elements, line elements, and surface elements. Some scholars evaluated the measurement of information and information distribution by defining and then implementing the measurement, calculating the measurement using test maps, and finally comparing the measurement values with human judgment on map information. A size measurement of an object-based Voronoi region (points, respectively) can be used to identify the distribution of information [12]. Some scholars have quantitatively analyzed the toponymy data, divided the toponymy information of Hubei province into mountain-related toponymy and water-related toponymy, visualized them, and finally calculated the traditional map information entropy [13]; however, the results found that the greater the information entropy, the more uniform the distribution of toponymy types, which should be more accurate to the research at rural scale.

Additional studies have focused on the spatial-temporal evolution of toponyms and cultural landscapes from the perspective of time. For example, scholars used methods including geographic information map, kernel density estimation, and correlation coefficient analysis, respectively, taking Northeast China, North China, and the Yangtze River plain as examples, to analyze the spatial and temporal evolution characteristics of the toponym cultural landscape in county-level administrative regions since the Sui Dynasty (581–619 CE) and to analyze the correlation among population density, digital elevation model (DEM), gross domestic product (GDP), and toponym density [14]. Other scholars have established the spatial-temporal database of the evolution of toponyms, set the spatial-temporal change index from the natural and cultural perspectives, and expressed the influence index in the form of a geographic information map to analyze the differences in the spatial distribution of toponyms density since the Qin Dynasty (221–207 BCE) [15]. From the perspective of spatial correlation, this study emphatically analyzed the difference and stability of spatial agglomeration of toponyms and cultural landscapes on both sides of the “Hu line”, which divided 96% of China’s population on the eastern side of the line [16] as well as the semantic evolution of toponyms and cultural landscapes. The above studies focused on the characteristics and internal mechanisms of toponym cultural landscape evolution in different time series and which evolution could be shown in map. However, the problem lies in that the division of county-level administrative regions has changed substantially from ancient to modern times. The current county-level administrative planning does not represent the county-level administrative regions in the previous dynasties. Therefore, if smaller and more stable rural-level units are used as research objects, more representative results will be obtained.

Further studies have focused only on the migration or distribution of ethnic groups. The use of toponym information in several studies reflects ethnic characteristics and changes in topography and land use. The principal component analysis (PCA) method is used to determine the past racial groups, using population data from the current dialects of contemporary ethnic groups, trend surface analysis, spatial clustering, and study of the minority populations using Kriging interpolation migration, as well as the historical evolution of the local landscape characteristics [17]. Previous scholars have studied the distribution and the structure of the Manchu eight banners for the major ethnic groups in Northeast China, while county-level toponyms are divided into those from Manchu and those from Manchu autonomous region, and the geographical distribution of each group and the spatial pattern of toponyms of Manchu, Mongol, and Han were divided by kernel density estimation [2]. However, these studies cannot combine the spatial distribution characteristics of ethnic groups with their migration patterns to study their commonality and characteristics, and the results are not precise enough due to the fact of their large scale and singular methodology.

By integrating the research progress of predecessors, the innovation of this study is reflected in that the data of toponyms at the level of rural settlements are selected for the first time for research, and the research results are more detailed and accurate. Furthermore, the study area was expanded into Northeast China, instead of being limited to the three northeastern provinces, and the results are more accurate and complete. The kernel density estimation and standard deviational ellipse method were used to analyze the spatial patterns, and the spatial distribution characteristics of ethnic groups were combined with their migration tendencies. The spatial information entropy was calculated based on geographical location information, including map, geometrical distribution, topological adjacency, and thematic information entropy. In addition, the spatial distribution characteristics of ethnic groups were quantitatively investigated from the perspective of information entropy.

## 2. Materials and Methods 

### 2.1. Study Area

Throughout Chinese history, the northeast region has been claimed by several different dynasties and tribes, and this is why the region now features many different kinds of ethnic groups [18,19]. Therefore, we cannot use the administrative boundary from only one single dynasty as the study area. This toponymy study area (Figure 1) includes all of Northeast China in the traditional sense, including Liaoning, Jilin, and Heilongjiang provinces in Northeast China and Inner Mongolia; and the outer region (i.e., Chifeng, Hulunbeier, Tongliao, and Xingan leagues). In addition, because Chengde, Qinhuangdao, and Tangshan (Hebei Province) have many areas and summer resorts associated with the Qing dynasty, these were also included in the study area. The ethnic groups in the region mainly include the Manchu, Mongol, Korean, and Xibe. The study area contains 11 autonomous Manchu counties including one in Jilin Province, six in Liaoning Province, and four in Hebei Province as well as a large number of autonomous Manchu townships.

Manchuria was originally named by Nurhachi, who lived in the city of Odori in the mohui of Russia in Eastern Changbai; thus, “Manchuria” is a tribal name. The name of three northeastern provinces “began in the Qing Dynasty (1636–1912 CE) [20]. After emperor Guangxu ascended the throne in 1875 CE, he divided Fengtian, Jilin, and Heilongjiang into three northeast provinces again”. In the 33rd year of the reign of Emperor Guangxu (1907 CE), the three northeastern provinces were transformed into provinces with governors and local governments [21]. In 1954 CE, the People’s Republic of China(PRC) changed its administrative divisions, and most of the western part of the three eastern provinces were divided into Inner Mongolia (the four eastern leagues region), reducing the size of the area considerably. 

### 2.2. Data Sources and Processing Methods

We developed a database of toponyms (including details of the origin and other characteristics) for Northeast China based on the National Database for Geographical Names of China [22], mainly including urban residential areas, rural residential areas, village committees, agricultural, forestry and pasture sites, industrial areas, public institutions, party, and government organizations. In addition, this database contains the Northeast China provinces, city, and county toponyms, the toponyms dictionary, statistical yearbook, and related historical documents. The borders of the county-level administrative divisions in Northeast China used in this paper were obtained from the county-level administrative boundaries provided by the National Geomatics Center of China [23]. Using database management software to extract keywords, it screened for China’s Northeast toponyms of various etymologies, avoiding many Manchu, Mongol, Korean in autonomous’ county and townships unrelated to Manchu, Mongol, Korean cultures of toponyms, and has extracted the real Manchu, Mongol, Korean, Hui, and Xibe related toponyms. 

According to the following sequence of Chinese historical dynasties: Jin (1115–1234 CE); Yuan (1271–1368 CE); Ming (1368–1644 CE); Later Jin (1616–1636 CE); Qing (1636–1912 CE); Warlord Conflict Period (1913–1930 CE); Manchukuo Period (1931–1945 CE); and Eve of the Founding of PRC (1946–1949 CE); we extracted the timeline showing when each rural settlement appeared for all rural settlements. We note that toponyms in the Jin, Yuan, Ming, Later Jin, and Qing dynasties were based on the extraction of key words related to these dynasties. For toponyms in different areas, there may be a small overlap in the timeline. Among them, the Qing Dynasty was the period of great prosperity of the Manchu, and the expansion of rural settlements in the research area was extremely obvious. Therefore, for the Qing Dynasty, the rural settlements were further subdivided into the Early (1636–1795 CE) and Late (1796–1912 CE) Qing dynasties according to the emperor’s year numbers. The Warlord Conflict Period was set at 1913 to 1930, the Manchukuo Period was set at 1931 to 1945, and the Eve of the Founding of PRC was set at 1946–1949. The classification of rural settlements by timeline is shown in Table 1.

Subsequently, ArcGIS software was used to process the data, and the toponyms of rural settlements located in different timelines were classified and displayed to observe more intuitively the expansion and migration of rural settlements in Northeast China. There are 609,507 toponymy points in the database, 83,320 rural settlements, and 13,721 ethnic toponyms were extracted. The statistical results are shown in Table 2.

After the projection (WGS_1984_Web_Mercator_Auxiliary_Sphere) conversion of the research area in ArcGIS, a grid size of 30 × 30 km was used to generate a grid, and a total of 3,277 grids were produced. We counted the number of toponymy points in each grid and the toponymy points related to ethnic groups, ranked and matched the color according to the number of toponymy points, filled the interior color of the grid according to the number of toponymy points in each grid, and filled the border color according to the number of toponymy points related to ethnic groups in each grid. The resulting diagram of layered color settings is shown in Figure 2.

### 2.3. Technical Processing Flow

The technical flow chart is shown in Figure 3. This study used GIS spatial analysis to calculate the spatial information entropy based on geographic information, including map information entropy, geometrical distribution, topological adjacency, and thematic map information. In general, these toponyms reflect China’s northeast Manchu and the related time and space distribution characteristics of the nation. Therefore, analyzing the spatial distribution of ethnic toponyms influenced by different factors and exploring the historical background of the relationship between people and the environment is intended to reveal the profound influence of the modern northeast immigration of Manchu culture and explain the national fusion and the evolution of the migration process. The spatial distribution of ethnic minority toponyms was generated using the kernel density estimation method, and a correlation analysis was performed with the range and direction distribution of standard deviational ellipse. The characteristics of temporal and spatial changes of rural settlements in Northeast China were analyzed as were the factors affecting rural settlements expansion and migration. The spatial distribution characteristics of toponyms and cultural landscapes were analyzed from the perspective of geographical regions, and the ancient and modern maps were compared to reflect the migration and cultural development of ethnic groups in Northeast China, and the historical changes of rural settlements were analyzed from the perspective of history, indicating the expansion and migration path of rural settlements. Through the outcome of the resulting map, the historical landscape and cultural landscape of Northeast China are visually presented.

### 2.4. Research Methods

#### 2.4.1. Kernel Density Estimation

As there are many data points in the study area and the spatial autocorrelation dominates the whole, a spatial interpolation model based on spatial autocorrelation was selected. With enough sample points, the error of the kernel density estimation (KDE) method was relatively small, and so it was deemed suitable to calculate the sample point density [24]. This method does not use the prior knowledge of data distribution and does not attach any assumptions to the data distribution; rather, it is a method for studying the characteristics of data distribution from the data samples themselves. Therefore, it is highly valued in the field of statistical theory and application. This method uses a smooth peak function (the “kernel”) to fit the observed data points to simulate the real probability distribution curve; it is a non-parametric method used to estimate the probability density function. According to the sample point group of a single variable, the estimated value of its spatial smoothness was calculated. The KDE has *n* sample points with independent and identical distribution *F*, and its probability density function, *f*, and can by calculated by Equation (1):(1)$$f_{h}\left(x\right)=\frac{1}{n}\overset{n}{\underset{i=1}{\sum }}K_{h}\left(x-x_{i}\right)=\frac{1}{nh}\overset{n}{\underset{i=1}{\sum }}K\left(\frac{x-x_{i}}{h}\right),$$
where *K(.)* is a pre-given inverted u-shaped function, called the kernel function; and *h > 0* is a smoothing parameter, called bandwidth, used to define the size of the smoothing quantity, which is actually a radius of a circle centered on x; and *K_h_*(*x*) = $\frac{1}{h}$
*K*($\frac{x}{h}$) is the scaled kernel.

Kernel density analysis can calculate the density of the element in its surrounding neighborhood and the point element density around each output grid pixel. Each point is covered with a smooth surface. The surface value is the highest at the point location and gradually decreases as the distance from the point increases [25]. The surface value is zero at the location where the distance from the point is equal to the search radius. The volume of space enclosed by the surface and the plane below is equal to the population field value of this point, and the density of each output grid pixel is the sum of the values of all the core surfaces superimposed on the grid pixel center.

#### 2.4.2. Standard Deviational Ellipse

The KDE mainly obtains the density distribution map of place points that have a distribution direction which also requires using the standard deviational ellipse (SDE) method, which can represent the direction and distribution of data [26]. The calculation principle uses the center of arithmetic mean to calculate the ellipse center. As shown in Equations (2) and (3), *X_i_* and *Y_i_* are the spatial position coordinates of each element, and *SDE_x_* and *SDE_y_* are the calculated ellipse centers. Then, the direction of the ellipse is determined taking the *x*-axis as the standard. Due north is 0 degrees. Finally, the length of the *x,y*-axis is determined, as shown in Equation (4):(2)$$SDE_{x}=\sqrt{\frac{\sum_{i=1}^{n}{\left(x_{i}-\bar{X}\right)}^{2}}{n}},SDE_{y}=\sqrt{\frac{\sum_{i=1}^{n}{\left(y_{i}-\bar{Y}\right)}^{2}}{n}},$$
(3)$$\tan\theta =\frac{A+B}{C},\;A=\overset{n}{\underset{i=1}{\sum }}{\bar{x}}_{i}^{2}-\overset{n}{\underset{i=1}{\sum }}{\bar{y}}_{i}^{2},\;B=\sqrt{{\left(\overset{n}{\underset{i=1}{\sum }}{\bar{x}}_{i}^{2}-\overset{n}{\underset{i=1}{\sum }}{\bar{y}}_{i}^{2}\right)}^{2}+4{\left(\overset{n}{\underset{i=1}{\sum }}\bar{x_{i}}\bar{y_{i}}\right)}^{2}},\;C=2\sum_{i=1}^{n}\bar{x_{i}}\bar{y_{i}},$$
(4)$$\sigma_{x}=\sqrt{2}\sqrt{\frac{\sum_{i=1}^{n}{\left(\bar{x_{i}}\cos\theta -\bar{y_{i}}\sin\theta \right)}^{2}}{n}},\sigma_{y}=\sqrt{2}\sqrt{\frac{\sum_{i=1}^{n}{\left(\bar{x_{i}}\sin\theta -\bar{y_{i}}\cos\theta \right)}^{2}}{n}},$$

The long and short half-axes of the ellipse represent the data distribution direction and range, respectively. The shorter the axis, the more obvious the centripetal force. The center point represents the center position of the full set. Generally, the center point position is roughly consistent with the position of the arithmetic average, as long as the data variability are not great. When elements have a normal spatial distribution (that is, they are most dense at the center and gradually become sparse as they approach the periphery), the first standard deviational (default value) range can include the center of mass of the input elements, accounting for approximately 68% of the total.

### 2.5. Information Entropy Calculation

#### 2.5.1. Map Information Entropy

Entropy is a quantitative measure of the amount of information contained in a message. Shannon (1948) defined the concept of information entropy based on the probability theory of mathematical statistics, which was used to represent the degree of uncertainty eliminated after obtaining a piece of information and to measure the size of information [27]. The more uncertain the message, the more information it contains. This method is statistically rigorous. Sukhov, a professor in the former Soviet Union, initially portrayed Shannon’s information entropy in cartography to assess map information [28]. *N* is the total number of symbols in the map, *M* is the total number of map symbol classes, the number of such symbols is *k_i_*, and the frequency of each symbol type is as in Equation (5). Taken in the information entropy calculation Equation (6), the map information of each type of symbols can be derived [29]:(5)$$P\left(x_{i}\right)=\frac{k_{i}}{N}\;i=1,2,\ldots ,M,$$
(6)$$H\left(x\right)=-\sum_{i}^{n}P\left(x_{i}\right)*log_{2}P\left(x_{i}\right),$$

This narrow sense of information entropy is only based on mathematical statistics and cannot reflect the spatial distribution relationships in maps, because spatial information is not only simple statistical and topological information but also may contain geometrical and thematic elements. When designing quantitative measures of spatial information, the spatial position and distribution of map symbols should also be considered. Therefore, Li and Huang [30] used the Voronoi region of symbols to model the map symbol spatial distribution and then established a new quantitative measurement method for map spatial information. With the Voronoi diagram, discrete map elements are shown organically, which advances the geometrical information [31] (including the position of map elements, the information, such as quantity, size, and shape), the topological information (including the topological map elements, direction, distance, and distribution of information), and project of information (including the type of map elements, importance degree, and other information).

Liu Huimin et al. [32] slightly improved Li and Huang’s geometrical information, reflecting the degree of uniform distribution continuity through the difference in point element distribution density, and then proposed geometrical distribution information and topological adjacency information. In addition, a Delaunay triangulation net was established based on the Voronoi adjacency relation of the point elements, and the edges of the triangulation net were subject to the statistical distance constraint from whole to local in turn, which interrupted the inconsistent edges in the triangulation net and obtained a series of subgraphs, each of which formed a spatial cluster, or cluster structure. By counting the number of point elements, distribution area, and distribution density of each cluster structure, the information of the cluster structure can be calculated.

#### 2.5.2. Information Entropy Calculation Method

Geometrical distribution Information. Map information is generated from the diversity and difference of spatial elements and their distribution features. The distribution map of toponyms and points in this study is a point-like map, whose geometrical distribution features are shown as the degree of aggregation, and distribution uniformity which can be reflected by the distribution density differences of the point elements [33]. This point element distribution density is determined by the relationship between the distance and direction of adjacent points and is inversely proportional to the area of the corresponding region. The larger the area, the sparser the element distribution, the fewer elements there are overall, and the less corresponding information there is. The larger the regional area difference, the more uneven the point distribution and the richer the distribution form, providing more information. Thus, it can be seen that the geometrical information generated by the geometrical distribution characteristics of point elements was negatively correlated and positively correlated with the regional area of point elements and the size of their differences, respectively.

When calculating the spatial information content of a point-shaped thematic map, first, a constraint diagram is constructed for the point elements of the thematic map to obtain the Voronoi region corresponding to each point element, and then the Voronoi region area corresponding to each point element is calculated. The Voronoi region area is used to calculate the geometrical distribution information content of the map. Because the Delaunay triangulation method used by the tool is most suitable for data in a projected coordinate system, the geographic coordinate system is first transformed into a projected coordinate system. The irregular triangulations (TINs) conforming to the Delaunay criterion were divided into all the points [34]. The perpendicular bisectors of the triangle sides form the sides of a Voronoi polygon. The intersection of the bisectors determines where the Voronoi polygon folds.

The area is taken as the description index of the geometrical distribution characteristics, and the feature-based information calculation model is substituted to obtain the expression of the geometrical distribution information of the point-shaped thematic map as *I_G_*, as shown in Equation (7), where *V_area_* corresponds to the element of the *i*-th point; and *V_Areai_* and *V_Areamax_* are the average and maximum values of the area. The units of information obtained by logarithm base 2 are bits. The geometrical distribution information is generated by the distribution density of point elements and its difference:(7)$$I_{G}=\overset{N}{\underset{i=1}{\sum }}\log_{2}\left(\frac{\left|V_{Area_{i}}-\bar{V_{Area}}\right|}{V_{Area_{max}}}+1\right).$$

Topological Adjacency Information. The topological adjacency information of the map is calculated by using Voronoi order adjacency as a description index. In terms of the topological adjacency characteristics of the spot thematic graph, the topological adjacency of the graph is the main adjacency relation, which is generally expressed by the topological adjacency degree. The degree of topological adjacency reflects the element connectivity, and the difference in adjacency reflects the difference in spatial connectivity. The larger the difference, the greater the topological adjacency information [35].The degree of adjacency selects the first-order topological adjacency degree of the Voronoi region corresponding to the point element as the description index of the topological adjacency information measurement of the point element, and normalizes it by adding the information model based on the feature to obtain the point map topological adjacency information generated by the topological adjacency feature, as shown in Equation (8), where, *t_i_* is the first-order adjacency of Voronoi region corresponding to the *i*-th point element:(8)$$I=\sum_{i=1}^{N}\log_{2}\left(\frac{\left|t_{i}-\bar{t}\right|}{t_{max}}+1\right),$$

Thematic Information. The thematic information of the map is related to the feature theme type. If all the neighbors [36] of a symbol are of the same theme type, the information of the symbol in the theme sense is very low [37]. However, if a symbol has a neighborhood of different theme types, it should be considered to have higher thematic information. We let the *i*-th map symbol have *N_i_* first-order adjacent symbols, which have *M_i_* theme types, where, the number of symbols of the *i*-th thematic type is *N_i_*. Then the frequency of the *i*-th neighbor symbol containing the *j*-th thematic attribute type is defined as Equation (9). The thematic information of the *i*-th map symbol is *H_i_ (TM)*, and the calculation formula is shown in Equation (10). If all the symbol neighbors are of the same theme type, the calculated result value is 1, so the thematic information is counted by 0:(9)$$P_{j}=\frac{n_{j}}{N_{j}},\;j=1,\;2,\;\ldots ,\;M_{i},$$
(10)$$H_{i}\left(TM\right)=\sum_{j=1}^{M_{j}}\log_{2}{\left(\frac{n_{j}}{N_{j}}+1\right)}_{,}$$

## 3. Results and Discussion 

### 3.1. Geographical Distribution of the Ethnic Toponyms

The KDE and SDE analysis results of the Manchu are shown in Figure 4a. Because there are a high number of toponyms related to Manchu in Northeast China, the distribution direction is not obvious. Therefore, rural settlements related only to Manchu were selected for analysis to better represent the directional characteristics of Manchu geographical distribution. The kernel density analysis of Manchu-related toponyms found that the highest density was located in the eastern part of Liaoning Province and the southern part of Jilin Province. The first SDE contained 68% of the data points, and the second SDE contained 95% of the data points. The analysis results of the two ellipses had the same direction which were distributed along the “wicker border” edge, which refers to the dyke ditches built in Northeast China in the second half of the 17th century. The Qing Dynasty (1636–1912 CE) saw the rise of the Manchu northeast as: “ancestral king’s house,” “dragon born land”. To protect the “benefits of the sacred region” in this area from being destroyed by the North Koreans and to prevent the invasion of the foreign vassal Mongolia, they dredged the border trenches in the northeast area and planted willows along the trenches, which are called the willow edge [38]. Manchu rural residential areas are more densely distributed, the highest density is in the inside of the willow edge, and then gradually spreads to the outside. Their settlements were concentrated in a large area; so, the Manchus were the dominant ethnic group in the study area.

The KDE and SDE analysis results of Mongol are shown in Figure 4b. The Mongol are typical herding nomadic people, mainly distributed in Inner Mongolia but without obvious directionality. The distribution is extremely concentrated, and the maximum kernel density is the largest among the five ethnic groups, so its distribution is the densest.

The KDE and SDE analysis results of the Korean are shown in Figure 4c. The toponyms of the Korean are mainly distributed in the border between Jilin Province, Liaoning Province, and North Korea. Geographically, each Korean community is relatively small but dense, and several Korean communities are distributed in the Manchu community.

The Hui’s KDE and SDE analysis results are shown in Figure 4d. Hui is distinguished from the previous three groups by the scattered distribution of the group across the study area. The Hui is external, and the overall distribution range is very wide but has no obvious distribution direction or expansion trend.

The KDE and SDE analysis results of the Xibe are shown in Figure 4e. The Xibe is a typical compact ethnic group in a small area, mainly distributed in Liaoning Province.

### 3.2. Expansion and Migration of Rural Settlements

According to the historical timeline, the core density of rural settlements between the former and the new dynasties was compared. If the density of the new dynasty was higher than that of the previous dynasty, it was regarded as the migration of the rural settlement center. In the legends of all the pictures in this section, the “Highest Density Region” represents the region with the largest kernel density according to the calculated results after applying kernel density analysis.

In 1115, the Jin Dynasty was founded as the capital of Huining prefecture (now Harbin, Heilongjiang Province), and its name was Dajin (1115–1234 CE). It was a feudal dynasty in Northern and Northeastern China, established by the Nvzhen. Through kernel density analysis, it can be concluded that the rural settlement center distribution was indeed located in Harbin and gradually spread to the southwest around the center. Rural settlements in the Yuan Dynasty (1271–1368 CE) were concentrated in Hebei Province and gradually spread to Liaoning. Compared with the Jin dynasty, it is evident that the rural settlements migrated, and the migration distance was large which was related to the fact that the Nvzhen were nomadic people at that time. Rural settlements in the Ming Dynasty (1368–1644 CE) were concentrated in Hebei Province, similar to those in the Yuan Dynasty but were larger in scope. In general, during the period from the Jin Dynasty (1115–1234 CE) to the Ming Dynasty (1368–1644 CE), it is evident that the rural settlements migrated southward as shown in Figure 5a.

In 1616, Nurhachi, as the founder of the Qing Dynasty, founded the Later Jin Dynasty (1616–1636 CE). His son Huangtaiji changed the country name to Qing Dynasty in 1636. According to the kernel density analysis, the rural settlements were mainly distributed in Liaoning, which is consistent with modern historical interpretations, because its capital was located in Hetuala (today’s Fushun city, Liaoning Province). The rural settlements migrated again, and the density center migrated to Liaoning Province. In the Shunzhi years (1638–1661 CE), rural settlements were concentrated in Liaoning Province, similar to that in the Later Jin Dynasty but with a larger central area. Therefore, from the Ming Dynasty (1368–1644 CE) to the Early Qing Dynasty (1638–1795 CE), the rural settlements migrated northward. The migration map is shown in Figure 5b.

Under the rule of 11 emperors in the Qing Dynasty, rural settlements in the research area continued to expand. During the reign of Jiaqing years (1796–1820 CE), a new center of the rural settlement began to focus on Jilin province. The possible reasons being that the rural settlement in Liaoning Province was too dense, it did not have much forest land available to the west for the Mongolian territory and to the south and eastern sea. Therefore, it could only gradually expand northward. In addition, the wicker border, which was originally the boundary between Manchu and other nationalities, was abandoned in the 10th year of Xianfeng (1860 CE), and a large amount of land was unappropriated but available for farming. Therefore, the overall scope of rural settlements in the Late Qing Dynasty (1796–1912 CE) expanded significantly, gradually forming a new rural settlement gathering center. The expansion range from Early Qing Dynasty to Late Qing Dynasty is shown in Figure 5c.

During the Warlord Conflict Period (1913–1930 CE), the Feng clan warlords mainly concentrated in Liaoning and Jilin; therefore, to avoid the war, the rural settlement center moved north to Heilongjiang Province again. The migration route from the Late Qing Dynasty (1796–1912 CE) to the Warlord Conflict Period (1913–1930 CE) is shown in Figure 5d.

Because the capital of the “Manchukuo” (the puppet Manchukuo was not recognized by the national government, the communist party of China, or the international community) was established in Changchun, Jilin Province, the rural settlement center moved south to Jilin Province. The migration route from the Warlord Conflict Period (1913–1930 CE) to Manchukuo Period (1931–1945 CE) is shown in Figure 5e.

Owing to the restoration of peace and stability during the founding of PRC (1949 CE), the rural settlements continued to move south to the more livable Liaoning Province. The migration route from the Manchukuo Period (1931–1945 CE) to the Eve of the Founding of PRC (1946–1949 CE) is shown in Figure 5f.

This study analyzed the spatial distribution characteristics of toponyms and cultural landscapes from the perspective of geographical regions and breaks through the conventional framework of expressing spatial changes by administrative zoning. In GIS, the spatial smoothing method based on KDE and spatial interpolation technique reflects the spatial changes of toponyms and landscapes. The selection of toponyms with historical significance can reflect the temporal change of toponyms over long periods. Through a series of time series analyses, we can clearly see the expansion and migration of rural settlements in northeast China. In the Jin (1115–1234 CE) and Yuan (1271–1368 CE) dynasties, rural settlements were scattered, with few large settlement areas. In the Ming Dynasty (1368–1644 CE), rural settlements were concentrated in Hebei Province, and settlement areas gradually took shape. In the Later Jin (1616–1636 CE) and the Early Qing Dynasty (1636–1795 CE), there was an obvious migration of the rural settlements, and the density center moved northward to Liaoning province. During the Jiaqing (1796–1820 CE) period, rural settlements began to shift significantly for the second time, and the newly formed rural settlement centers began to be concentrated in Jilin Province. A possible reason for this was that rural settlements in Liaoning were already too dense, and there was not much land left for settling; therefore, they began to expand northward. During the Warlord conflict period (1913–1930 CE), the rural settlement scope expanded obviously, and the rural settlement center moved northward again. In the Manchukuo Period (1931–1945 CE), the rural settlement centers began to move back to southward due to the establishment of the Manchukuo. This trend continued until the Eve of the Founding of PRC (1946–1949 CE), referring to the historical population data set of the History Database of the Global Environment (HYDE) [39] as the evidence of population migration and using the 1950 CE rural data set as shown in Figure 6. 

The data in the study area were divided into four levels: Zero to indicate no residents; Low, few residents; Medium, residents starting to gather with medium density; and High, large number of residents gathering with high density. As can be seen from Figure 6, up to 1950, the densely populated areas in the study area were Liaoning province and the central part of Jilin province, showing a relatively obvious banded distribution, which was consistent with the migration results.

The size of each grid in Figure 6 is 1 × 1 km, and the total population of the covered grids in each provincial administrative region is counted and compared with the number of toponyms in rural settlements, as shown in Table 3. It can be seen that there is a certain correlation between population and the total number of rural settlements.

### 3.3. Correlation Analysis based on Information Entropy Theory 

#### 3.3.1. Map Information Entropy

Traditional map information entropy is calculated for the toponymy data in the study area, which is more traditional, but ignores the specific form of map information transmission. The calculated results based on Equations (5) and (6) are shown in Table 4. The map information entropy (denoted as *I_M_*) in Manchu toponyms account for the largest proportion, followed by Mongol and Korean. The larger the *I_M_* value, the more ethnic toponyms in the region. The results show that the Manchu were mainly distributed in Liaoning, Jilin and Heilongjiang provinces. The Mongol were mainly distributed in Inner Mongolia, while the Korean were mainly distributed in Jilin and Heilongjiang provinces. The Hui are evenly distributed in all provinces, while Xibe are mainly distributed in Liaoning Province. The calculation of map information entropy is only able to obtain the location information of toponyms and cannot analyze the density of toponyms.

It is also meaningful to study the number of rural settlements with respect to timeline. The I_M_ of each dynasty and period are calculated based on Equations (5) and (6), and the results are shown in Table 5. The *I_M_* in Qing Dynasty accounted for the largest proportion, followed by Warlord Conflict Period and Ming Dynasty. This is consistent with the statistics in Section 2.2. The provinces of *I_M_* showed that most of the rural settlements in Liaoning, Jilin, and Inner Mongolia were formed during the Qing dynasty. Rural settlements in Heilongjiang were mainly formed during the Qing Dynasty and the Warlord Conflict Period, while those in Hebei were mainly formed during the Ming and Qing dynasties.

#### 3.3.2. Geometrical Distribution Information

The toponyms of the Manchu, Mongol, Korean, Hui, and Xibe were generated into Voronoi polygons and are shown in Figure 7. The area of the polygon of each point located in the pair was counted, and the geometrical distribution information entropy (denoted as *I_G_*) was calculated based on Equation (7) as shown in Table 6. The larger the regional area difference, the more uneven the point distribution and the richer the distribution form, thus, the more information. Therefore, it can be seen that Mongolian toponyms have the most obvious clustering characteristics, followed by Manchu, while the Hui, Xibe, and Korean have the same clustering results as KDE result.

It is also meaningful to study the aggregation of toponyms with respect to timeline. Building Voronoi polygons for each dynasty and period, the number of rural settlements and the calculated results of *I_G_* based on Equation (7) are shown in Table 7. The larger the area difference of each region, the more uneven the distribution of point position, which means, the richer the distribution form, so as the greater the information contained. The rural settlements in the Qing Dynasty accounted for 75.1% of the total rural settlements, and the geometrical distribution information contained in the Voronoi polygon was much larger than that in other dynasties and periods. Therefore, the distribution and aggregation characteristics of the Qing Dynasty were the most obvious indices and the overall trend of expansion. The geometrical distribution information contained in the Warlord conflict period, the Manchukuo Period, and the Eve of the Founding of PRC are similar to each other, indicating that the density of rural settlements in the region has tended to be stable at that time, but the mass migration was continuing due to the war and political policies. It can be seen that the calculation result of *I_G_* was the same as the KDE result.

#### 3.3.3. Topological Adjacency Information

The degree of topology adjacency reflects the connectivity among elements, and the difference in adjacency reflects the difference in spatial connectivity. The calculation results of topology adjacency information entropy (denoted as *I_Topo_*) based on Equation (8) are shown in Table 8. It can be seen that the adjacency of Mongol toponyms is the most obvious, followed by that of Manchu. Different from the calculated results of geometrical information distribution, the adjacency of the Korean is significantly different from that of the Hui and the Xibe, which is the same as the calculated results of the SDE.

#### 3.3.4. Thematic Information

In this study, a constraint diagram was constructed for the toponymic points, and the Voronoi region corresponding to each point was obtained. For each polygon in the Voronoi region, the thematic information entropy contained in it was calculated according to Equation (11). The *N_j_* is the total number of polygon neighbors, and *n_j_* is the number of neighbors of the polygon’s theme type.
(11)$$I_{j}=\log_{2}\frac{n_{j}}{N_{j}},$$

Therefore, when all neighbors of a polygon are polygons of the same theme type, the thematic information is 0. When the *N_j_* value is the same, the smaller the *n_j_* value is, the more important the theme meaning contained in the polygon is, while the *I_j_* value is smaller, that is, the *I_j_* value of a polygon is negatively correlated with the information of the theme meaning contained in the polygon. The hierarchical colormap of thematic information calculations for each polygon in the Voronoi region is shown in Figure 7.

It can be clearly seen from the figure that the border area between Heilongjiang and Hulunbeier has a high thematic information entropy, and it can be inferred that this is the place where Mongolian and Manchu rural settlements meet, as well as the place where cultural integration occurs. On the contrary, the Inner Mongolia region, where Mongolian and Manchu gather, the wicker edge coastal region has low thematic significance; so, it can be inferred that they gather to a high degree and have low uncertainty.

In order to display the results of the thematic information more intuitively, all Voronoi polygons were counted according to the city area. The research area of this study was large, so it was feasible to calculate by city area. If the research area was small, it could also be calculated by county area or town area. By calculating the sum of the thematic information contained in all Voronoi polygons in each city, the diagram of layered color setting by the sum of the thematic information in the city map of the study area was shown in Figure 8. The greater the thematic information, the more obvious the phenomenon of multi-ethnic intermingling in this area, and the smaller the thematic information, the more obvious the phenomenon of single-ethnic settlement.

If all the neighbors of a symbol are of the same theme type, then the symbol is of very little importance in the thematic significance. However, if a symbol has a neighborhood of different theme types, it should be considered to have higher thematic information. The calculated thematic information (denoted as *I_Th_*) results of all ethnic toponyms based on Equations (9) and (10) are shown in Table 9.

The calculated results of geometrical distribution information and topological adjacency information were the largest for the Mongol, but the calculated results of thematic information show that the Manchu was the largest, so there were more residential areas of other nationalities around the rural residential areas of Manchu which had a higher thematic significance. The Mongol was so densely distributed that its thematic information decreased, while the Korean’s thematic information increased.

## 4. Conclusions 

Overall, the results demonstrated that, as a symbol language, toponyms have dynamically recorded changes in the human environment over its long historical development process and inherited the unique local historical culture. According to the spatial distribution of toponyms and the calculation results of spatial information entropy, it can be seen that Manchu occupied a dominant position among the ethnic groups in the study area, and the region with the largest kernel density was located in the eastern part of Liaoning Province and the southern part of Jilin Province, showing an obvious linear distribution. Mongolian is mainly distributed in Inner Mongolia, which is denser but less extensive than Manchu and has no obvious direction. The Korean is mainly distributed in the border region of Jilin, Liaoning, and North Korea, and its settlement scope is relatively small but also dense. As a kind of immigrant group, the Hui is scattered throughout the research area. The Xibe is a typical compact ethnic group in a small area, mainly distributed in Liaoning Province. According to the spatial distribution of rural settlements, the research results of rural settlement migration were consistent with the history of the study area. From the Ming Dynasty to the Warlord Conflict Period, there has been a continuous northward migration. After the Manchukuo Period and the Eve of the Founding of PRC, the rural settlement center began to return to the south, and there was a certain correlation between the density of toponyms and population density. 

The results of this study are conducive to the protection and promotion of toponym culture and improves our understanding of the unique cultural information contained in toponyms. In the meantime, a large number of geographical data extracted from this study can be published on the China Historical Geographic Information System (CHGIS) to supplement the missing information in border areas and aid in solving problems of insufficient spatial coverage. Toponyms are repositories of the interaction between human history and natural change, with historical, philosophical, and scientific value. The toponyms remaining from previous dynasties should not be changed at will, and we should cherish and protect these ancient and meaningful toponyms. Toponyms are repositories of the interaction between human historical processes and natural changes, with historical, philosophical, and scientific values. In this study, the spatial information entropy is calculated based on the location information of toponyms, and the distribution and migration trajectory of ethnic groups and rural settlements can be inferred by combining the density and population information of toponyms. Of course, the distribution of topography, hydrology, and vegetation in the study area may also have certain influences on the distribution and migration of rural settlements. However, due to the space limitation and experiences, these parameters will be further investigated in the future. Moreover, this method can be applied to investigating a variety of spatial distribution phenomena and is likely to be applicable to other regions and ethnic groups, however we primarily encourage further study of this specific application.

## Figures and Tables

**Figure 1 entropy-22-00393-f001:**
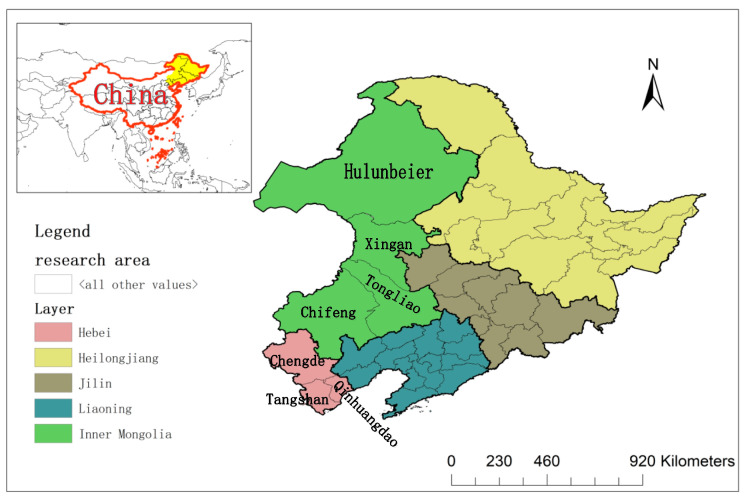
Study area.

**Figure 2 entropy-22-00393-f002:**
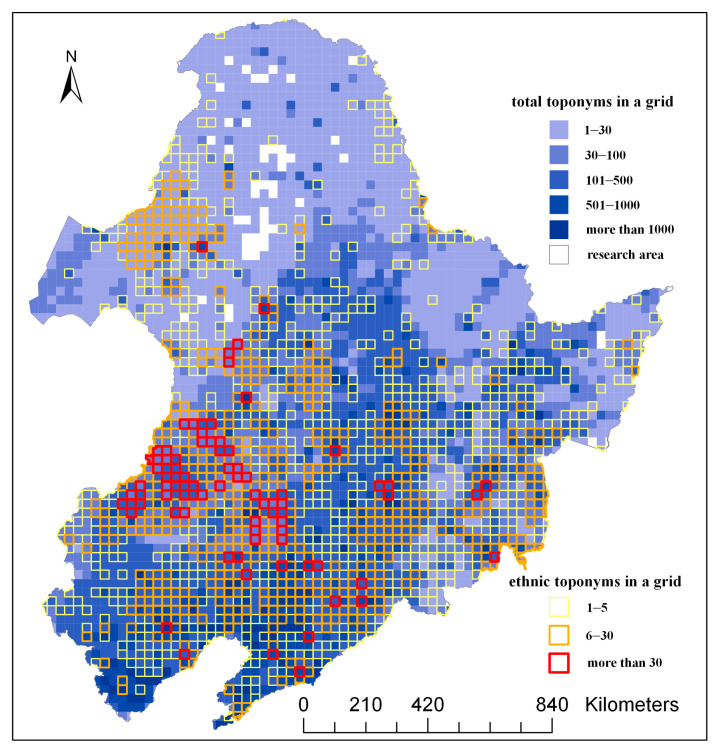
Raster data distribution diagram.

**Figure 3 entropy-22-00393-f003:**
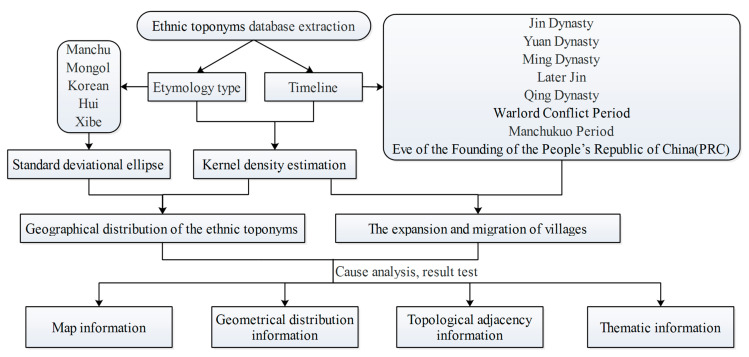
Technical process flow chart.

**Figure 4 entropy-22-00393-f004:**
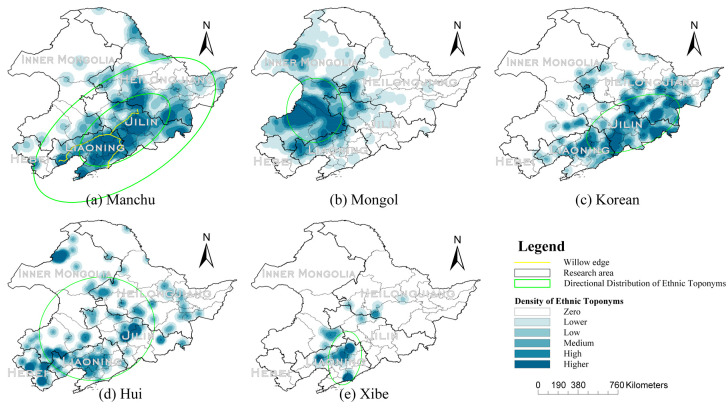
Geographical distribution characteristics of the ethnic toponyms in Northeast China: (**a**) Manchu toponymic layer; (**b**) Mongol toponymic layer; (**c**) Korean toponymic layer; (**d**) Hui toponymic layer; and (**e**) Xibe toponymic layer.

**Figure 5 entropy-22-00393-f005:**
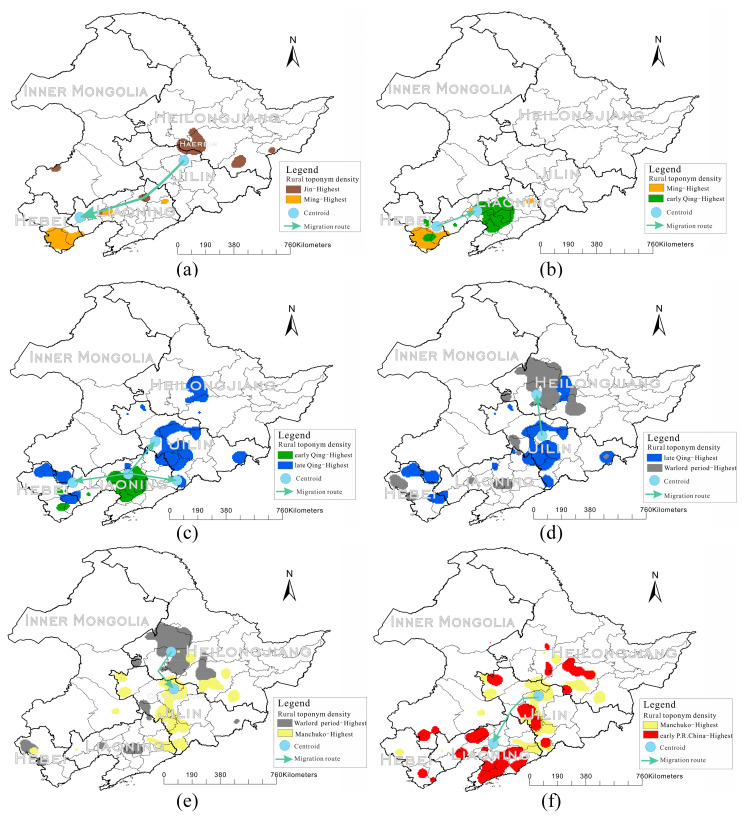
Maps of expansion and migration of rural settlements in Northeast China: (**a**) Jin-Ming migration; (**b**) Ming–Early Qing migration; (**c**) Early Qing–Late Qing expansion; (**d**) Late Qing–Warlord Conflict Period migration; **(e)** Warlord Conflict Period–Manchukuo Period migration; and **(f)** Manchukuo Period–Eve of the Founding of PRC migration.

**Figure 6 entropy-22-00393-f006:**
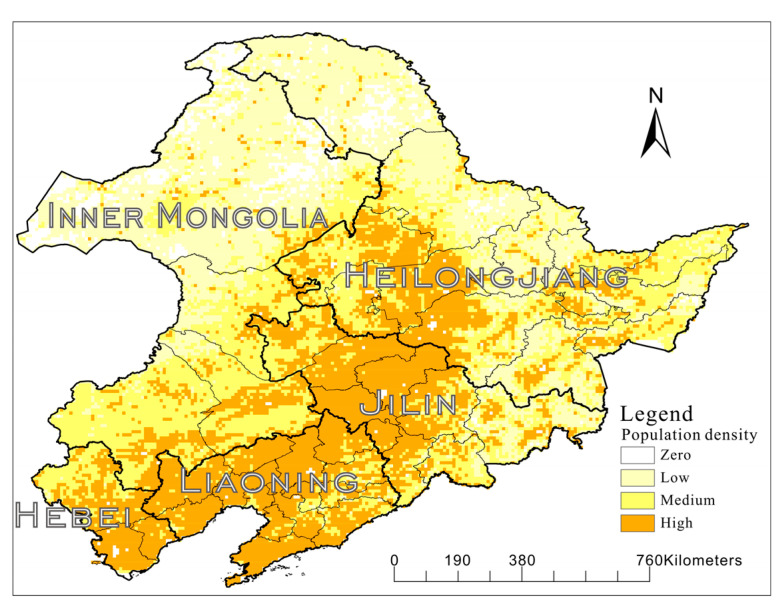
Rural Population density of Northeast China in 1950.

**Figure 7 entropy-22-00393-f007:**
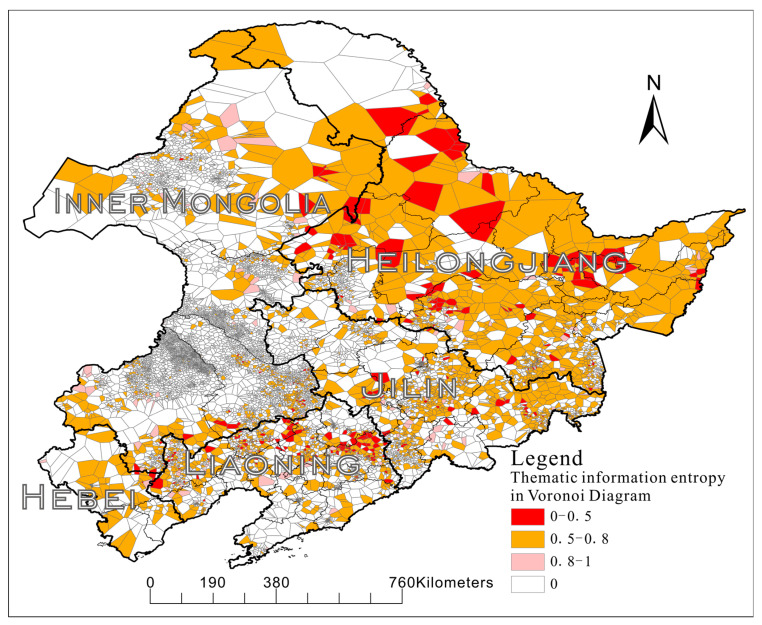
Voronoi polygon thematic information hierarchical colormap.

**Figure 8 entropy-22-00393-f008:**
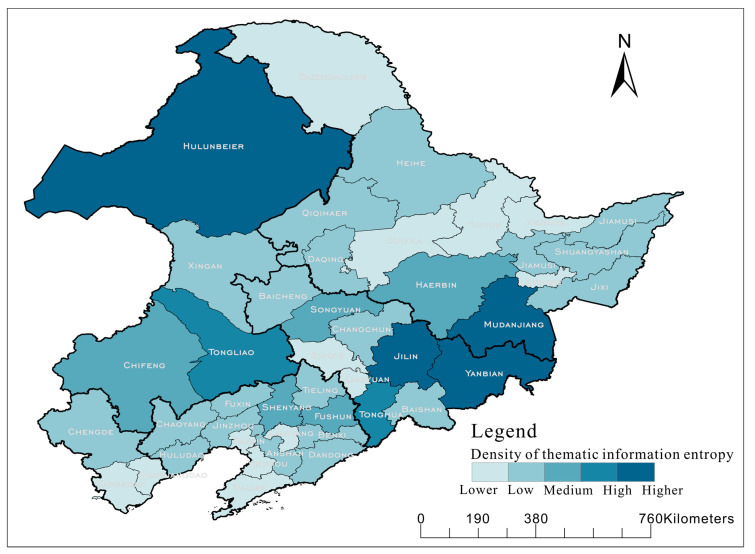
Ethnic toponyms thematic information distribution diagram.

**Table 1 entropy-22-00393-t001:** Timeline of rural settlements.

Time Period	Rural Settlements	Rural Settlements Ratio
Jin	147	0.2%
Yuan	424	0.5%
Ming	5915	7.1%
Later Jin	172	0.2%
Qing	62,555	75.1%
Warlord Conflict Period	6553	7.9%
Manchukuo Period	3943	4.7%
Eve of the Founding of PRC	3611	4.3%
Total	83320	-

**Table 2 entropy-22-00393-t002:** Statistical table of toponyms in the study area.

Location	Ethnic Toponyms	Rural Settlement Toponyms	Cumulative Toponyms
Liaoning	2,007	241,041	223,223
Jilin	1,996	25,422	108,867
Heilongjiang	1,705	12,122	131,057
Inner Mongolia	7,709	8,937	80,393
Hebei	136	12,798	65,967
Total	13,553	83,320	609,507

**Table 3 entropy-22-00393-t003:** Comparison of population and rural settlements in the study area.

Region	Population	Population Ratio	Rural Settlements	Rural Settlements Ratio
Liaoning	15,550,899	34.5%	72,324	32.1%
Jilin	11,153,415	24.8%	47,221	20.9%
Heilongjiang	8,076,123	17.9%	43,782	19.4%
Inner Mongolia	4,801,020	10.7%	25,618	11.3%
Hebei	5,464,403	12.1%	36,838	16.3%
Total	45,045,861	-	225,783	,

**Table 4 entropy-22-00393-t004:** Ethnic toponyms map information entropy.

Region	Manchu	Mongol	Korean	Hui	Xibe
I_M_	0.498237	0.418961	0.278584	0.169976	0.099483
Liaoning	0.248285	0.224337	0.079235	0.039238	0.093829
Jilin	0.271068	0.224337	0.159244	0.039238	0.003461
Heilongjiang	0.253551	0.154794	0.144851	0.023772	0.001878
Inner Mongolia	0.197072	0.492744	0.011482	0.053914	0.007677
Hebei	0.040594	0	0	0.036021	0

**Table 5 entropy-22-00393-t005:** Rural settlements map information entropy.

Region	Jin	Yuan	Ming	Later Jin	Qing	Warlord Conflict Period	Manchukuo Period	Eve of the Founding of PRC
*I_M_*	0.016137	0.038769	0.270918	0.018414	0.310476	0.288517	0.208285	0.196247
Liaoning	0.002886	0.011047	0.111745	0.01321	0.488729	0.062259	0.045285	0.109045
Jilin	0.001137	0.000991	0.018414	0.001838	0.506677	0.066176	0.123323	0.052611
Heilongjiang	0.010341	0.003264	0.005062	0.004946	0.259966	0.222453	0.084885	0.070808
Inner Mongolia	0.002107	0.012822	0.010645	0.000368	0.327042	0.006088	0.025727	0.029947
Hebei	0.002759	0.019926	0.209278	0.000842	0.317943	0.048368	0.016506	0.019749

**Table 6 entropy-22-00393-t006:** Ethnic toponyms geometrical distribution information.

	Manchu	Mongol	Korean	Hui	Xibe
*I_G_*	51.2885255	96.34024	28.36428	24.73101	24.27911
Ethnic Toponyms	3331	8517	1002	475	228

**Table 7 entropy-22-00393-t007:** Rural settlements geometrical distribution information.

	Jin	Yuan	Ming	Later Jin	Qing	Warlord Conflict Period	Manchukuo Period	Eve of the Founding of PRC
*I_G_*	23.37757	11.74949	29.37864	11.94902	64.70554	17.95237	17.02269	16.61073
Rural Settlements	147	424	5915	172	62555	6553	3943	3611

**Table 8 entropy-22-00393-t008:** Ethnic toponyms topological adjacency information.

	Manchu	Mongol	Korean	Hui	Xibe
*I_Topo_*	407.9045	863.8214	113.3573	56.18125	29.59878
Ethnic Toponyms	3331	8517	1002	475	228

**Table 9 entropy-22-00393-t009:** Ethnic toponyms thematic information.

	Manchu	Mongol	Korean	Hui	Xibe
*I_Th_*	1040.459	616.1091	486.2956	160.5137	66.70275
Ethnic Toponyms	3331	8517	1002	475	228

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
