# Peer review of "Spatial-Temporal Characteristic Analysis of Ethnic Toponyms Based on Spatial Information Entropy at the Rural Level in Northeast China"

_entropy, 2020, doi:10.3390/e22040393_

Round 1
Reviewer 1 Report
This paper represents an interesting application of geospatial and information metrics to the diffusion of toponyms in Northeast China. This relates to deeper philosophical questions—particularly those raised by the French Marxist Henri Lefebvre—of how a place’s identities develop through the social production of space. That said, the paper is very much an empirical and technical study, and devotes little attention to deeper reflection. This is not to disparage the quality work that the authors have done, but to suggest to them that they could increase the significance of and interest in their study by more explicitly linking it to theoretical discussions in historical geography and the use of Historical Geographic Information Systems (HGIS). I am not intimately familiar with this literature, but a simple web search indicates that Fudan University is working with Harvard University on a China Historical Geographic Information System (https://sites.fas.harvard.edu/~chgis/pages/intro/). The authors may wish to consult this work to consider how their study could contribute to this project.
My statistics are a bit rusty, but as best as I can tell, the study is solid and the methodology appropriate to the research objectives. Thus, most of my suggestions are for relatively minor changes to the way the information is presented.
Although minor grammatical errors are present and occasionally create ambiguities, the overall mastery of English language and style is more than sufficient to adequately convey the authors’ key arguments.
For readers not familiar with Chinese history, it would be helpful to mention the times of the dynasties alongside their names. For example, on line 58: “… since the Sui dynasty (581-618 CE), and …”
Likewise with the Hu Line, e.g. (line 65): “… both sides of the “Hu line” dividing China into two roughly equal parts, as well as the …”
Considering how integral the maps are to the presentation of the results, these could be improved considerably.
Particularly useful for international readers would be an initial map identifying the provinces in the study area. Once I got to the Materials and Methods, I found such a map. On lines 88-89, the authors should probably refer readers to this map, e.g.: “The study area was expanded to Northeast China, not limited to the three northeastern provinces (see figure 14) …” In this and all other maps where they appear, the province borders should be rendered darker than the county borders to indicate the distinction.
Other countries and the rest of China could be included in the maps, but shaded out to indicate that they do not fall within the study region. At the very least, the Korean peninsula should be included, since diffusion of toponyms from there is part of the study.
According to the explanation on lines 299-316, the contemporary provinces and counties do not correspond to historical territories. Thus, the authors may want to reconsider including contemporary provinces and counties in their historical maps (figures 6-11), so as to avoid cartographic anachrony.
The map legends should be cleaned up and given more descriptive labels. For instance, in figure 6, remove the KernelID_Ming dynasty and KernelID_Jin dynasty subheadings, and change the labels for these features to Ming dynasty: highest density region and Jin dynasty: highest density region. The “research area” subheading is also unnecessary. The text referring to the figure should explain what is meant by “highest density region.” Also, placing the chronologically prior category above the subsequent one in the legends would be more intuitive.
What map projection was used?
Since one of the distinguishing points of the study is the use of finer-scale data (lines 86-88), it may be illustrative to overlay the provincial and county maps onto the Voronoi diagram.
Going back to the broader significance of the study in the conclusion, on lines 531-533, the authors mention that “Place names are repositories of the interaction between human history and natural change...” which is true, but raises an issue with respect to the subsequent (lines 538-539) contention that “migration trajectory of ethnic minorities can be inferred by combining the density and population information of place names.” Specifically, would not factors such as topography, hydrology, and vegetation also be important to consider here? This is not to say that the authors’ claim is not valid, but that they may want to be a bit more specific regarding the extent to which their study actually engages with the complexities of the nature-society dialectic.
Author Response
Dear Reviewer:
Thank you for your letter and for the reviewer comments concerning our manuscript entitled “Spatial-temporal Characteristic Analysis of Ethnic Toponyms Based on Spatial Information Entropy at Rural-Level in Northeast China” (ID: entropy-739585). We thank you for your appreciation of our contributions. We also thank you for your detailed comments that helped us improve the quality of this paper. Those comments are very valuable and also be much helpful for revising and improving our paper, as well as the important guidance signifying to our researches. Revised portion are marked in highlight in the paper. The main corrections in the paper and the responds to the reviewer comments are as flowing.
Sincerely,
Zhiqiang Xie
Comment 1: They could increase the significance of and interest in their study by more explicitly linking it to theoretical discussions in historical geography and the use of Historical Geographic Information Systems (HGIS). The authors may wish to consult this work to consider how their study could contribute to this project.
Response 1: This study examined the CHGIS data, and found that its spatial scope was not sufficient to cover the study area. On the other hand, there were only a few years in the border areas where some data existed, which was not enough to support the study's demand for the timeline. There is also no detailed information on each site in CHGIS to examine its historical significance. In conclusion part, we explained that the results of this research can make a certain contribution to CHGIS. (lines 605-608)
Comment 2: For readers not familiar with Chinese history, it would be helpful to mention the times of the dynasties alongside their names.
Response 2: We agree with your suggestion and added the times of the dynasties alongside their names. And briefly explains the meaning of “Hu line” (line81).
Comment 3: Particularly useful for international readers would be an initial map identifying the provinces in the study area. In this and all other maps where they appear, the province borders should be rendered darker than the county borders to indicate the distinction.
Response 3: We agree with your suggestion and annotations were added. The quality of all figures in the paper has been improved, the provincial boundaries were bolded to facilitate the reader's distinction between provinces and cities.
Comment 4: Other countries and the rest of China could be included in the maps, but shaded out to indicate that they do not fall within the study region. At the very least, the Korean peninsula should be included, since diffusion of toponyms from there is part of the study.
Response 4: We agree with your suggestion and have fixed figure 1(study area, line146).There are many Manchu in China outside the study area, but those areas are not the birthplace of Manchu. What we do is mainly to study the Manchu changes in northeast China through toponyms, rather than the national migration. Due to languages and ethnic reasons, some of the Manchu people may have migrated to the Korean peninsula, and they may not have Manchu marks on the toponyms, which is beyond the scope of this study. In the further research, it is expected to invite linguists and historians to discuss the temporal and spatial changes of the Manchus in the Korean peninsula and beyond, and we hope to further cooperate with you to study this topic.
Comment 5: The contemporary provinces and counties do not correspond to historical territories. Thus, the authors may want to reconsider including contemporary provinces and counties in their historical maps, so as to avoid cartographic anachrony.
Response 5: We are sorry that this part was not clear in the original manuscript. In 2.1 we added the reason to choose to use contemporary provinces instead of historical territories. (lines115-118)
Comment 6: The map legends should be cleaned up and given more descriptive labels. The text referring to the figure should explain what is meant by “highest density region.” Also, placing the chronologically prior category above the subsequent one in the legends would be more intuitive.
Response 6: We agree with your suggestion and have modified some map legends. In addition, the explanation of “highest density region” was added in 3.2. (lines399-401)
Comment 7: What map projection was used?
Response 7: Thank you very much for pointing out our negligence. The map projection we used was WGS_1984_Web_Mercator_Auxiliary_Sphere projection. (line187)
Comment 8: Since one of the distinguishing points of the study is the use of finer-scale data, it may be illustrative to overlay the provincial and county maps onto the Voronoi diagram.
Response 8: We agree with your suggestion and added the provincial and civic maps onto the Voronoi diagram (Figure 7). Too many boundary lines in the county map will affect the reader's view of the polygons in the Voronoi diagram, thus were not added.
Comment 9: Specifically, would not factors such as topography, hydrology, and vegetation also be important to consider here?
Response 9: We agree with your viewpoint. The distribution of topography, hydrology and vegetation in the study area will also have certain influences on the distribution and migration of rural settlements. However, due to space limitation and experiences, these issues will be further studied in the future. We have added these sentences to the conclusion part. (lines 616-618)
Reviewer 2 Report
- The term “standard deviation ellipse” should be “standard deviational ellipse”.
- The keywords are not the same as the words used in the abstract and context, e.g. geographical place pattern, Manchu culture, rural settlements migration.
- It is not clear about the data of the study area in section 2. How many rural settlements in different dynasties are collected for the analysis? And please give a brief introduction to the data resource and the base maps.
- It will be much better if the important place names and administrative names are annotated in the figures.
- The structure of this paper is unreasonable. The section2 and section 3 should be exchanged.
- The kernel density methods have been used to study the culture distribution patterns for many years. Is there a novel kernel density proposed in this paper?
- A few information entropies are used in this paper. How did they help us to analyze the spatial-temporal characteristics of ethnic toponyms?
- There are many more references related to this paper, but only 25 materials are listed in the reference section.
Author Response
Dear Reviewer:
Thank you for your letter and for the reviewer comments concerning our manuscript entitled “Spatial-temporal Characteristic Analysis of Ethnic Toponyms Based on Spatial Information Entropy at Rural-Level in Northeast China” (ID: entropy-739585). We also thank you for your detailed comments that helped us improve the quality of this paper. Those comments are very valuable and also be much helpful for revising and improving our paper, as well as the important guidance signifying to our researches. Revised portion are marked in highlight in the paper. The main corrections in the paper and the responds to the reviewer comments are as follows.
Sincerely,
Zhiqiang Xie
Comment 1: The term “standard deviation ellipse” should be “standard deviational ellipse”.
Response 1: We agree with your suggestion and have changed all “standard deviation ellipse” mentioned in this study to “standard deviational ellipse”.
Comment 2: The keywords are not the same as the words used in the abstract and context, e.g. geographical place pattern, Manchu culture, rural settlements migration.
Response 2: Thank you very much for pointing out our negligence. We have modified the keywords, changed the “geographical name” involved in this study to “toponym”, and unified the meaning of “rural settlements”.
Comment 3: It is not clear about the data of the study area in section 2. How many rural settlements in different dynasties are collected for the analysis? And please give a brief introduction to the data resource and the base maps.
Response 3: We are sorry that this part was not clear in the original manuscript. We added a few explanations on the data we use. We have added a table of rural settlements in each dynasty (Table1). The introduction to the data resource and the base maps are in 2.2 (lines 149-157). We corrected a table (Table 4) in which the number of rural settlements was used instead of the number of all toponyms.
Comment 4: It will be much better if the important place names and administrative names are annotated in the figures.
Response 4: We agree with your suggestion and annotations have been added to the maps. The quality of all figures in the paper has been improved.
Comment 5: The structure of this paper is unreasonable. The section2 and section 3 should be exchanged.
Response 5: We agree with your suggestion and have exchanged the section2 and section3. We also updated the numbers of all the figures, tables, and equations in the paper.
Comment 6: The kernel density methods have been used to study the culture distribution patterns for many years. Is there a novel kernel density proposed in this paper?
Response 6: We are sorry that this part was not clear in the original manuscript. The focus of this study is not on how to improve the algorithm but on the analysis of kernel density results. Kernel density estimation is a classical method to calculate density curve. Its analysis principle and results are applicable to the data in this study and can be used for further study of expansion and migration.
Comment 7: A few information entropies are used in this paper. How did they help us to analyze the spatial-temporal characteristics of ethnic toponyms?
Response 7: We are sorry that this part was not clear in the original manuscript. We should have explained that in 3.3, we have revised the contents of this part. When we calculate the distribution of ethnic minorities, we use ethnic toponyms in our database. When we calculate the expansion and migration of rural settlements along with the timeline, we use the rural settlement data in our database. The calculation results of the map information, geometrically distributed information, topological information and thematic information entropy are all meaningful to determine the spatial characteristics of ethnic toponyms. But considering the temporal characteristics of rural settlements, only the calculation results of the map information and geometrically distributed information entropy are meaningful.
Comment 8: There are many more references related to this paper, but only 25 materials are listed in the reference section.
Response 8: We agree with your suggestion and have added more references.
Round 2
Reviewer 2 Report
No.
Author Response
Dear Reviewer:
Thank you for your letter and for the reviewer comments concerning our manuscript entitled “Spatial-temporal Characteristic Analysis of Ethnic Toponyms Based on Spatial Information Entropy at Rural-level in Northeast China” (ID: entropy-739585). We also thank you for your detailed comments that helped us improve the quality of this paper. Those comments are very valuable and also be much helpful for revising and improving our paper, as well as the important guidance signifying to our researches. The manuscript has been rechecked and the necessary changes have been made in accordance with the reviewers’ suggestions. The responses to all comments have been prepared and attached herewith/given below.
Sincerely,
Zhiqiang Xie
Comment 1: English language and style are fine/minor spell check required .
Response 1:
- We have polished the manuscript by the Editage (www.editage.cn) for English language editing. As a result, the details of the content of the paper have been slightly changed.
- The manuscript has been rechecked and the necessary changes have been made including the format of references,figures and tables. And the content of the paper has been slightly modified and deleted, which further elaborates on the meaning of the paper. For example, we abbreviate some commonly methods or terrms and delete the incomprehensible paragraphs that used to introduce Manchu language to facilitate the reader's understanding.